# Utilizing the DNA Aptamer to Determine Lethal α-Amanitin in Mushroom Samples and Urine by Magnetic Bead-ELISA (MELISA)

**DOI:** 10.3390/molecules27020538

**Published:** 2022-01-15

**Authors:** Jiale Gao, Nuoya Liu, Xiaomeng Zhang, En Yang, Yuzhu Song, Jinyang Zhang, Qinqin Han

**Affiliations:** Faculty of Life Science and Technology, Kunming University of Science and Technology, Kunming 650500, China; jialegaojl@163.com (J.G.); lny15387658750@163.com (N.L.); Z1303167982@163.com (X.Z.); yangen@kust.edu.cn (E.Y.); syzzam@126.com (Y.S.); zhangjinyangzjy@163.com (J.Z.)

**Keywords:** α-amanitin, aptamer, truncated aptamers, MELISA, mushroom, urine

## Abstract

Amanita poisoning is one of the most deadly types of mushroom poisoning. α-Amanitin is the main lethal toxin in amanita, and the human-lethal dose is about 0.1 mg/kg. Most of the commonly used detection techniques for α-amanitin require expensive instruments. In this study, the α-amanitin aptamer was selected as the research object, and the stem-loop structure of the original aptamer was not damaged by truncating the redundant bases, in order to improve the affinity and specificity of the aptamer. The specificity and affinity of the truncated aptamers were determined using isothermal titration calorimetry (ITC) and gold nanoparticles (AuNPs), and the affinity and specificity of the aptamers decreased after truncation. Therefore, the original aptamer was selected to establish a simple and specific magnetic bead-based enzyme linked immunoassay (MELISA) method for α-amanitin. The detection limit was 0.369 μg/mL, while, in mushroom it was 0.372 μg/mL and in urine 0.337 μg/mL. Recovery studies were performed by spiking urine and mushroom samples with α-amanitin, and these confirmed the desirable accuracy and practical applicability of our method. The α-amanitin and aptamer recognition sites and binding pockets were investigated in an in vitro molecular docking environment, and the main binding bases of both were T3, G4, C5, T6, T7, C67, and A68. This study truncated the α-amanitin aptamer and proposes a method of detecting α-amanitin.

## 1. Introduction

Mushroom poisoning is a global problem, which poses a threat to human health. Mushroom poisoning occurs worldwide, due to accidental picking and consumption [1]. One study reported that 90% of deaths from mushroom poisoning were caused by mushrooms containing amatoxin [2]. α-Amanitin is the most abundant lethal toxin among the amatoxins, with a lethal dose to humans of about 0.1 mg/kg [3]. When wild mushrooms containing amatoxin are accidentally consumed, the amatoxin enters the hepatocytes and binds to RNA polymerase II, which inhibits the transcription of mRNA and protein synthesis, eventually leading to necrosis of hepatocytes. In addition, amatoxin promotes the expression of apoptosis factor caspase-3, Annexin V, and the P53 protein, and decreases expression of the anti-apoptosis factor Bcl-2 protein, causing damage; however, the exact mechanism remains unclear [4,5,6]. No specific antidote has been found for amatoxin. It has been reported that polymyxin B competes for the RNA polymerase II binding site; thus, inhibiting the binding of α-amanitin to RNA polymerase II. Animal experiments have shown that polymyxin B significantly reduces the liver and kidney injury induced by α-amanitin in mice [7], but no further clinical trials have been reported.

α-Amanitin is a cyclic octapeptide. Cyclic peptides have a richer structural and functional diversity and better conformational and metabolic stability, due to their sequence diversity and loop structure, compared with linear polypeptides [8]. Owing to their specific structures, cyclic peptides have become a hot research topic in biology and pharmacology. According to toxicological studies, amatoxins disappear rapidly from the serum, but can be detected in the urine within 4 days after ingestion [9,10]. As urine samples are relatively easy to obtain and the duration of amatoxins in urine is longer than that in serum, urine seems to be an obvious sample matrix for rapid detection of amatoxins. In recent years, more assays have emerged to detect α-amanitin. For example, instrumental analyses have a high specificity and sensitivity, but these methods require expensive instrumentation and complex sample pre-treatment [11,12,13,14]. Staak [15] was the first to use an enzyme-linked immunosorbent assay (ELISA) to detect amatoxin in urine. He [16] prepared a monoclonal antibody that specifically binds to amatoxin and established a method for detecting amatoxin in mushrooms using an indirect competitive immunoassay, which identified α-amanitin, β-amanitin, and γ-amanitin, with detection limits of 4.55, 4.9, and 4.45 ng/mL, respectively. Although ELISA is simple and highly sensitive, the method has the disadvantages of high cost and long cycle time to prepare the monoclonal antibodies (mAbs), which need further improvement. Investigators have further established the lateral flow immunoassay (LFIA) to overcome the drawbacks of ELISA. Bever [17,18] applied prepared mAbs to immunoassay strips to detect amatoxin. That experiment showed that LFIA can detect amatoxin in mushroom samples and urine, and the entire assay can be completed in 10 min, resulting in the detection of a large number of samples in a short time. However, mAbs have a long preparation cycle, making aptamers a better choice for detecting α-amanitin.

Aptamers are single-stranded DNA or RNA molecules that fulfill particular functions by coiling and folding into specific tertiary structures in the absence of the complementary strand and are screened from random oligonucleotide libraries using systematic evolution of ligands by exponential enrichment (SELEX), for highly sensitive and target-specific binding of the aptamers [19]. Aptamers undergo conformational changes and three-dimensional (3D) folding through base complementary pairing and other molecular forces, forming relatively stable 3D structures, such as hairpins, stem-loops, and G-quartets, which are the basis for aptamer-target binding [20,21]. Aptamers are also known as chemical antibodies, because their target recognition pattern is very similar to that of antigens and antibodies. However, aptamers have many advantages over antibodies. For example, aptamers have a low molecular weight, high stability, are easy to modify, have a wide target recognition range, and lack immunogenicity [22]. Therefore, aptamers have become important research tools in the fields of analytical chemistry and biochemistry. A biosensor based on the polymerase chain reaction (PCR) and streptavidin as a dual fluorescence polarization (FP) amplifier has been used to detect chloramphenicol residues in food, with a linear detection range of 0.001–200 nM [23]. An aptamer labeled with a reporter group (fluorophore) was used for qualitative or quantitative analysis of a test target by detecting the change in the signal upon binding of the aptamer to the target. By labeling the 5′ end of the aptamer with carboxy fluorescein and the 3′ end with Black hole quencher 1, the established aptamer molecular beacon method could be used to rapidly detect Aflatoxin B1 (AFB1) in samples [24]. The researchers also used the Thioflavin T (ThT) substitution assay, in which the ThT molecule fluoresces when bound to G4-quadruplex DNA, and the change in signal can be measured. FenA2 interacts with ThT to produce a label-free detection mode with a Kd of 33.57 nM (9.30 ppb), and the detection limit for Fenitrothion was 14 nM (3.88 ppb) [25].

The initial length of an aptamer is usually 70–100 nt, and not all bases are involved in the binding of the aptamer to the target. The excess bases may form a large spatial block, which hinders the recognition and binding of the aptamer to the target and increases the cost of synthesis [26]. In addition, a primitive aptamer with a long oligonucleotide sequence binds its target analog via a primer region or additional fragment, leading to impaired specificity [27]. A shorter aptamer has better penetration ability and facilitates the design and synthesis of aptamer probes for practical applications. Truncating the 88-nt melamine aptamer to 34-nt results in comparable binding ability and greater specificity to the target [27]. The affinity increased significantly and the Kd value of the aptamer decreased from 174 ± 27 pmol/L to 14 ± 1 pmol/L after truncating the 77-nt electric eel acetylcholinesterase aptamer to 39-nt [28]. However, when the specific platelet-derived growth factor adaptor was truncated to 36-nt, the affinity of the sequence increased 150 times, and the effect on the corresponding fibroblasts was consistent with the original sequence [29]. In summary, it is necessary to optimize the sequence after the target aptamer is screened.

Isothermal titration calorimetry (ITC) is a method that can be used to directly measure the thermal changes of reactions between substances and to directly obtain the parameters of the intermolecular binding process [30,31]. Through an ITC experiment, the dissociation constants (affinity, Kd), reaction heat (ΔH), and the number of binding sites (ratio of ligand to macromolecule, N) can be obtained easily and quickly. Gold nanoparticles (AuNPs) are a nanoscale material with a high extinction coefficient and diameters of 1–100 nm [32]. AuNPs dispersed in solution appear red. When the appropriate concentrations of salts (NaCl), surfactants, or cationic polymers are added, the color of the AuNP solution gradually changes from red to purple or blue, due to the aggregation of AuNPs [33]. Negatively charged single-stranded DNA (ssDNA) aptamers retain the stability of the AuNPs even in highly saline solutions, and the color of the solution remains burgundy. However, in the presence of a target, the aptamer binds to the target, and the appropriate NaCl concentration causes polymerization of the AuNPs, and the color of the solution changes from red to purple-blue [34,35]. This combination of AuNPs and an aptamer can be used for colorimetric detection of a target and to identify the affinity and specificity of the aptamer and target. ELISA is now widely used in various laboratories due to its simple operation, lack of need for large instruments, and simple sample pre-treatment process [36]. The combination of ELISA technology and magnetic beads for target detection is an emerging strategy. Magnetic beads are composite materials consisting of small metal particles containing superparamagnetic properties. When a magnetic field is present, the beads adsorb the magnetic field and separate from their solution; they return to their dispersed state when the magnetic field disappears [37,38].

In this study, we propose a colorimetric biosensor based on AuNPs and isothermal titration calorimetry (ITC) to identify the affinity and specificity of aptamers. The principle is that, after specific binding of the inducer to the target, the AuNPs aggregate and change color from red to purple when they encounter the salt solution. We developed a magnetic bead-based enzyme linked immunoassay (MELISA) to detect α-amanitin, and this represents the first use of aptamer for aα-amanitin detection. In this study, the aptamer was selected to replace the traditional antibody, and the biotin-labeled aptamer was immobilized on streptavidin magnetic beads using the streptavidin-biotin reaction to obtain magnetic beads that specifically recognized α-amanitin. The competition between the target, α-amanitin, and the complementary chain of the aptamer was used for detection, and a sensitive and rapid α-amanitin detection method was established (Figure 1).

## 2. Results and Discussion

### 2.1. Truncation and Sequence Design of the Aptamers

The stem-loop structure of the aptamer (also called the hairpin structure) is usually regarded as the site of recognition and binding to the target molecule. Therefore, as shown in Figure 2, the original sequence of the α-amanitin aptamer fitted using Mfold had two stem-loop structures, and the redundant arms at both ends of the aptamer were truncated without destroying the original aptamer stem-loop structure. Twelve tailored and optimized aptamers were obtained, and five sequences with the optimal structure of each tailoring method were selected by secondary structural prediction, as shown in Table 1. Among these, Apt-2 is obtained by cutting through 5′, Apt-8 is obtained by cutting through 3′ end, Apt-10 and Apt-11 are obtained by cutting through both 5′ and 3′ ends at the same time, and Apt-12 is obtained by cutting only one ring at both ends.

### 2.2. Specificity and Affinity Testing of the Aptamer

It has been shown that truncating to remove the excess aptamer bases improves the affinity of the aptamer. Truncated 38-nt and 12-nt aptamers specific for BPA were obtained through rational truncation from the 63-nt BPA aptamer. The dissociation constants (Kd) of the 38-nt and 12-nt aptamers were 13.17 nM and 27.05 nM [40]. Compared with the 66-nt aptamer, the 38-nt aptamer was shorter and had a higher sensitivity, better selectivity, and shorter detection time than 8-hydroxyguanine [41]. The secondary structure was predicted with Mfold, and the Gibbs free energy values of the six sequences are shown in Table 1. The affinity constants of the six sequences were analyzed (Figure 3A). As the aptamer concentration increased, protection of the AuNPs by the aptamer was enhanced. When the AuNPs encountered NaCl, the AuNPs maintained the original red color, and the higher the affinity, the more obvious the colloidal gold coagulation. Aptamer affinity was analyzed using GraphPad Prism 8 software, and the Kd values of Apt, Apt-2, Apt-8, Apt-10, Apt-11, and Apt-12 were 33.6 ± 3.928, 37.9 ± 6.053, 130.9 ± 31.1, 108.7 ± 18.23, 184.4 ± 37.08, and 132.1 ± 32.99 nM, respectively. The results showed that the affinity constants of the original sequences were the smallest, and the affinity constants of the truncated sequences were all higher than those of the original sequences. Previous studies have shown that the lower the affinity constant, the higher the affinity; therefore, the affinity of the truncated sequence was lower than that of the original sequence.

Isothermal titration calorimetry (ITC) is an effective method for characterizing aptamer binding [42]. The aptamer was gradually titrated into α-amanitin, and the amount of heat release was recorded as a function of time. With the addition of aptamer, the binding of aptamer and α-amanitin gradually tends to saturate. Therefore, the amplitude of heat change will gradually become smaller and the heat signal will gradually weaken, and the final titration curve will show an “S” curve. The results are shown in Figure 3C–I (SourceData in Appendix A). There was no caloric change in the control group, and the truncated aptamer had a different binding ability to α-amanitin. With the help of MicroCal PEAQ-ITC Analysis Software, the Kd values, number of binding sites (N), and each thermodynamic reaction parameter were calculated, as shown in Table 2. In conclusion, the original aptamer had desirable affinity and the results were consistent with the AuNPs assay.

As the color changed from red to violet-blue, the absorbance of the AuNPs at 520 nm decreased and the absorbance of the new peak at 620 nm increased. The absorbance values at 620 nm and 520 nm represent the state of the AuNPs [43]. Therefore, A620/A520 was used to evaluate the specificity of the aptamer to the target. The larger the A620/A520 value, the higher the specificity of the aptamer. The results of specificity analysis are shown in Figure 3B. The truncated sequence cross-reacted with non-specific targets and had no specific recognition for α-amanitin. The original sequence did not cross-react with non-specific targets, and did not accurately identify α-amanitin or β-amanitin.

In conclusion, the affinity and specificity of the aptamers decreased after truncation, so the original sequence was selected for subsequent research. The aptamer of CAP was removed 20 bases from the right primer region, to obtain the aptamer LR20, consisting of 40 bases, which has a higher affinity than L20; therefore, the aptamer primer region does not participate in the binding process [44]. Moreover, the deletion of the 7 bp region of the FiPA6-ThT stem segment, the FiPA6B-ThT sensor, showed a similar fluorescent signal, indicating the importance of this loop for the stem FiPA6 aptamer [45]. Previous studies have also shown that the binding affinity of the aptamer to AFB1 requires an appropriate number of base pairs in the stem region, which contributes to the formation of a stable stem-loop structure [46]. Therefore, the region of the aptamer that binds to the target is located in the stem-loop structure. Furthermore, molecular docking was used to simulate the binding sites of aptamers to α-amanitin, and the results are shown in Figure 4 for T3, G4, C5, T6, T7, C67, and A68, where the binding sites are located at both ends, while the optimized aptamer sequence by means of truncation at both ends is not suitable for all aptamers. Thus, the binding region of the aptamer to α-amanitin is located on both terminal arms and at the stem end adjacent to the ring, providing new ideas for subsequent studies of aptamer truncation.

### 2.3. Optimization of the Experimental Conditions

The aptamer concentration directly affected the α-amanitin capture concentration. Figure 5A (SourceData in Appendix A)shows that when the aptamer concentration was increased, the amount of aptamer immobilized on the magnetic beads increased. At this point, the absorbance value at 450 nm increased continuously until the aptamer concentration reached 100 nM, indicating that the aptamer concentration saturation was reached when the quantitative α-amanitin, complementary chain, and SA-HRP were added. Therefore, the optimum concentration of aptamer was chosen as 100 nM.

This experiment was based on the competition between the complementary chains of the aptamer and the α-amanitin, to achieve quantitative detection of α-amanitin. The more complementary chains bound to the aptamer, the higher the absorbance value at 450 nm. The concentration of the complementary chain was optimized in the experiment. As shown in Figure 5B, when the concentration of the aptamer complementary chain was 0–30 nM, the absorbance value of the solution at 450 nm increased continuously, and when the concentration reached 20 nM, the absorbance value reached saturation; thus, 20 nM was chosen as the optimum concentration for the aptamer complimentary chain.

### 2.4. Detecting α-Amanitin with the Aptamer-Based MELISA

To test the specificity of the MELISA, the structural analogs β-amanitin, γ-amanitin, phallacidin, and phallisacin were selected as negative controls, at a final concentration of 500 ng/mL, and skimmed milk was selected as the blank. The mixture included the following five targets: α-amanitin, β-amanitin, γ-amanitin, phallacidin, and phallisacin. The absorbance values of the other targets were compared to the blank. (Figure 5C). However, as the structures of β-amanitin and α-amanitin are very similar, and the difference in structural formula of the two is only the difference between -NH2 and -OH on the R3 arm, this method specifically detected α-amanitin compared to other biotoxins. This method can be used to specifically detect α-amanitin, but cannot accurately distinguish α-amanitin from β-amanitin. However, α-amanitin and β-amanitin are almost always present in mushrooms at comparable concentrations, so both analytes are biomarkers for detecting mushroom toxin poisoning [47]. Subsequent studies should be conducted to obtain specific aptamers for α-amanitin and to accurately differentiate α-amanitin from β-amanitin by reverse systematic evolution of ligands, by exponential enrichment to improve aptamer specificity.

The sensitivity of the assay was evaluated using the optimized conditions, and a series of standard α-amanitin solutions at different concentrations were measured. As the concentration of α-amanitin was increased, α-amanitin competed for more aptamers, resulting in fewer complementary chains bound to the aptamer and a decrease in the absorbance value of the solution at 450 nm. As shown in Figure 6A (SourceData in Appendix A), the absorbance value at 450 nm showed a linear relationship with the α-amanitin concentration in the range of 0.1–6 μg/mL, and the calibration curve equation was y = −0.2708x + 1.8725 (R^2^ = 0.99). This method was able to detect α-amanitin at a minimum of 0.1 μg/mL. Therefore, the limit of detection of the method was 0.1 μg/mL.

Compared with traditional ELISA [15], the magnetic bead-based enzyme linked immunoassay(MELISA) saves time and cost by immobilizing the aptamer and capturing the target with the help of magnetic beads. In addition, this method is highly sensitive and specific, does not require professionals to operate, and provides a reference for detecting other toxins. This method utilizes the targeted recognition function of the aptamer to achieve specific detection of α-amanitin, and the combination of streptavidin and biotin greatly improved the immobilization efficiency of the aptamer. In addition, the use of streptavidin magnetic beads allowed for separation and enrichment, making the method convenient. The detection limit of the magnetic bead-based enzyme linked immunoassay (MELISA) needs to be improved compared to the monoclonal antibody-based method for detecting amatoxin [16,17,18]. However, monoclonal antibodies take a long time to prepare and require high upfront costs. Therefore, the aptamer was used to replace the antibody, to establish a new method for detecting amatoxin, which greatly saves time and cost.

### 2.5. Detecting α-Amanitin in Actual Samples

The components of the test sample other than the analyte are called matrices, which often affect and interfere with the accuracy of the results when the sample is tested, and this is a major problem when testing actual samples. To assess matrix effects, actual samples of mushroom and urine were spiked and standard curves were established; the limit of detection in mushroom was 0.372 μg/mL and in urine was 0.337 μg/mL. In general, the closer the standard curve of α-amanitin in the actual sample is to that of the standard, the smaller the matrix effect it produces and the higher the recovery. As shown in Figure 6A,B, the mushroom and urine samples were close to the standard curve of α-amanitin. Furthermore, as shown in Table 3, the spiked recoveries of α-amanitin in mushroom was 84–95% and 83–96% in urine using commercial kits, and the spiked recoveries of α-amanitin in mushroom was 85–97% and 86–96% in urine using MELISA. The samples (mushroom leachate and urine) were tested using skim milk as the blank control, to further verify the feasibility of the method. The results are shown in Figure 6C and Table 4. The absorbance values of the mushroom leachate and patient urine samples at 450 nm were significantly lower (*p* < 0.05) than those of the blank control, and the results were consistent with HPLC, demonstrating that the method can detect α-amanitin in actual samples.

The traditional methods for detecting amatoxin are instrumental assays and liquid chromatography mass spectrometer (LC-MS). ELISA detects analytes in the ng/mL range [48,49,50]. The use of LC-MS requires complex sample pretreatment and expensive instrumentation. Previous studies have reported the development of a competition-based lateral flow immunoassay (LFIA) for the rapid, portable, selective, and sensitive detection of amatoxin, where extraction and detection can be accomplished in approximately 10 min [17,18]. This assay can be easily observed by eye and does not require instrumental reading. However, the LFIA is susceptible to false-positive and false-negative results due to storage conditions and matrix effects.

Due to the rapid metabolism of amatoxin in urine, many methods cannot be used to detect amatoxin in urine; therefore, a rapid method for detecting amatoxin in urine samples is needed. The MELISA is not affected by the sample matrix effect and can detect α-amatoxin in mushroom samples and urine, providing ideas for the development of clinical detection technology.

### 2.6. Molecular Docking Simulation Results

To investigate the interactive mode between α-amanitin and the aptamer, we predicted and analyzed the binding sites using molecular docking technology. The mutual recognition and specific non-covalent binding of the aptamer to α-amanitin occurred by hydrophobic interactions, van der Waals forces, and hydrogen bonding, and the minimum binding energy of α-amanitin molecule with aptamer was −4.26 kal/M. The results are shown in Figure 4. When α-amanitin bound specifically to the aptamer, it did not interact with all base sites of the aptamer, but only with the base sites at specific positions, such as T3, G4, C5, T6, T7, C67, and A68 of the single-stranded oligonucleotide. The hydrogen atoms on the α-amanitin ligand formed hydrogen bonds with oxygen atoms on the base of the T3 aptamer with a length of 2.2 A, while the amino hydrogen atoms formed hydrogen bonds with oxygen atoms in the skeleton between T6 and T7, with a length of 1.8 A.

Molecular simulations, to study the interactions between ligand and receptor molecules, help to understand the recognition and binding mechanisms, while providing a reference for subsequent mechanistic studies. Molecular simulation experiments are an auxiliary tool to study the binding sites between aptamers and targets, elucidate the structures of the aptamer-target complexes, and determine the binding pockets of the aptamers and targets, to more accurately describe the binding mechanism.

## 3. Conclusions

In this study, an aptamer was truncated to remove the redundant arms of the original α-amanitin aptamer, retain the loop structure, and truncate it into five different sequences. The affinity and specificity of the original aptamer and the truncated aptamer were identified. The results showed that the affinity and specificity of aptamers decreased after truncation. The aptamer affinity and specificity identification and molecular docking predicted the binding site of the aptamer to α-amanitin, indicating that the binding region of the aptamer to α-amanitin is located on the arm and at the stem end adjacent to the ring. Therefore, a novel method for detecting α-amanitin was established, using the original aptamer. An aptamer-based MELISA was developed, based on the traditional ELISA technique. The absorbance value at 450 nm was linearly related to the concentration in the range of 0.1–6 μg/mL, and the detection limit was 0.369 μg/mL, while in mushroom it was 0.372 μg/mL, and in urine 0.337 μg/mL. The use of magnetic beads made the method more sensitive, less time-consuming, and more efficient than traditional ELISA. Actual samples (mushroom leachate and urine) were tested under the optimal experimental conditions, and the results were as expected and consistent with those of HPLC.

## 4. Materials and Methods

### 4.1. Reagents

DON, OTA, AFB1, FB1, HAuCl4, and sodium citrate were purchased from Sigma-Aldrich Co. (St. Louis, MO, USA). α-Amanitin, β-amanitin, γ-amanitin, phallacidin, and phallisacin were obtained from the College of Life Sciences, Hunan Normal University (Changsha, China). Streptavidin-labeled magnetic beads were purchased from Primag Biotechnology Co., Ltd. (Xiamen, China). α-amanitin ELISA Kit was purchased from Shenzhen ziker Biological Technology Co., Ltd. (Shenzhen, China). Streptavidin-labeled horseradish peroxidase and the TMB color solution were purchased from TaKaRa (Beijing, China). Aptamer sequences, as shown in Table 1, were synthesized by Beijing Optimus Biotechnology Co., Ltd. (Beijing, China).

### 4.2. Preparation and Identification of the AuNPs

The AuNP solution was prepared using the sodium citrate reduction method [51]. First, 99 mL of deionized water and 1 mL of 1% chloroauric acid solution were added to a pre-soaked, dried, and treated 250 mL clean conical flask, and all solutions were filtered using an ultrafiltration membrane (0.45 μm). Subsequently, the solution was heated with a magnetic stirrer until it started to boil. Then, 1.5 mL of 1% sodium citrate solution was added quickly, the temperature of the magnetic stirrer was adjusted appropriately, and the color of the solution changed from light yellow to black, then from black to purple-red, and finally from purple-red to red. The solution was transparent. Finally, when the color of the solution no longer changed, the heating and stirring were continued for 10 min, and the conical flask was placed at room temperature and cooled. After cooling, the solution was dispensed and transferred to 4 °C for storage. The size, shape, and homogeneity of the colloidal gold particles were characterized by transmission electron microscopy and scanning the 400–600 nm UV-vis absorption spectra. Figure 7 shows that the prepared colloidal gold solution had a single absorption peak at the detection wavelength of 520 nm. No spurious peaks appeared, indicating that the prepared colloidal gold solution was homogeneous and dispersed.

### 4.3. Truncation of the Original α-Amanitin Aptamer Sequence

All truncated α-amanitin aptamers were truncated and designed based on the secondary structure of the original full-length 70-mer

(5′-CATGCTTCCCCAGGGAGATGGAGGTCTTTTTGGTTGTTGGTGGGGGAATCTTTTGGTATTGAGGAACATG-3′), as reported by Muszyńska et al. [39]. Excessively long aptamer sequences with large spatial site resistance easily affect the binding of the aptamer to the target, and an excessively long aptamer is not conducive to design and practical applications, which increases the synthesis cost [52]. Therefore, in this study, the secondary structure of the α-amanitin aptamer was simulated using Mfold software online (http://www.unafold.org/mfold/applications/dna-folding-form.php, accessed on 14 May 2021), and the 5′ end was trimmed, the 3′ end was trimmed, and both ends were trimmed simultaneously on the aptamer separately [53]. The two-end simultaneous trimming was performed by removing the extra two arms based on the original sequence stem-loop structure and then trimming the base pairs until the stem-loop structure was destroyed. Single-end trimming was performed in 3-base pairs, until the stem-loop structure of the original sequence was destroyed.

### 4.4. Aptamer Specificity and Affinity Tests

Different concentrations of the aptamer (0, 50, 100, 200, 400, 800, and 1600 nM) were incubated with 1 μg/mL of α-amanitin for 30 min at room temperature. AuNPs were added and incubated for 30 min at room temperature, followed by the addition of 1 M NaCl, to make a final concentration of 40 mM [54]. Absorbance values at 520 nm were measured using a UV spectrophotometer (Thermo BIOMATE3S; Thermo Scientific, Waltham, MA, USA). The aptamer concentration was used as the horizontal coordinate and (A′ − A_0_)/A_0_ was the vertical coordinate, and the aptamer affinity constants were calculated using GraphPad Prism 8.0 software (GraphPad Software Inc., La Jolla, CA, USA). A′ represented the A520 nm value of the aptamer at each concentration. A_0_ represented the A520 nm value when the aptamer concentration was zero.

Isothermal titration calorimetry (ITC) was conducted on a VPITC microcalorimeter. All solutions were degassed to avoid air bubbles. The α-amanitin was diluted with ultrapure water to a concentration of 41.6 μM. Experiment temperature: 25 °C, speed: 700 rpm, a total of 20 drops, the first drop: 0.2 μL, the other drops: 2 uL, each drop interval: 120 s. The titration phase: aptamers, concentration: 500 μM (50 μL), and the titrated phase: α-amanitin, concentration: 41.6 μM (350 μL). Set up a blank control: ultrapure water titrated with α-amanitin.

α-Amanitin, β-amanitin, DON, AFB1, FB1, and OTA (1 μg/mL) were incubated with 400 nM of the aptamer for 30 min at room temperature. AuNPs were added and incubated for 30 min at room temperature protected from light, and then 1 M NaCl was added to make a final concentration of 40 mM [54]. Absorbance values at 520 nm and 620 nm were measured with a UV spectrophotometer (Thermo BIOMATE3S) (A520 nm and A620 nm. A′: α-amanitin, β-amanitin, DON, AFB_1_, FB_1_, and OTA at a concentration of 1 μg/mL (A620 nm/A520 nm) and A_0_′: (A620 nm/A520 nm) values without a target. The graph was plotted with (A′ − A_0_′)/A_0_′ as the horizontal coordinate and the type of target as the vertical coordinate.

### 4.5. Optimizing the MELISA Experimental Conditions

Determining the optimal concentration of the aptamer was a key part of this experiment, as too low an aptamer concentration would lead to insignificant changes in absorbance values, and too high an aptamer concentration would lead to higher costs. First, the magnetic beads were cleaned twice with a magnetic bead combined buffer, and the volume in each tube after the last cleaning was 50 μL. The magnetic beads were resuscitated with a fit combination buffer of 3.125, 6.25, 12.5, 25, 50, 100, 200, and 400 nM, respectively, and gently shaken at room temperature for 15 min. The supernatant was discarded from the coupling-adapted streptavidin magnetic beads in the 1.5 mL centrifuge tube. Skim milk was added at 37 °C, and the tubes were closed for 2 h. A 100 μL aliquot of α-amanitin (final concentration of 500 ng/mL) was added to each tube and incubated at 37 °C for 2 h. A 100 μL aliquot of the aptamer-labeled complementary chain (20 nM) with a biotin label was added and incubated for 1 h at 37 °C. A 100 µL portion of 1000-fold diluted HRP was added to each tube and incubated at 37 °C for 1 h. The tubes were washed four times with PBST solution, after each of the above steps. A 100 μL aliquot of TMB color development solution was added and incubated at 37 °C for 15 min in the dark. After color development was completed, 50 μL of 2 M H2SO4 was added to each tube to terminate the reaction, and the supernatant was aspirated. The absorbance value of the supernatant solution was measured at 450 nm within 10 min of terminating the reaction using an enzyme marker.

The complementary chain concentration was optimized as follows: The steps were the same as described above to optimize the aptamer concentration, except for the first and fourth steps. First, the aptamer (final concentration 100 nM) was added. A 100 μL aliquot of different concentrations of the complementary chain solutions (0, 5, 10, 15, 20, 25, and 30 nM) was added to each tube. All other conditions were the same as above.

### 4.6. Detecting α-Amanitin Using the Aptamer-Based MELISA

The specificity of an assay is key to verifying the feasibility of the method. Therefore β-amanitin and γ-amanitin were the structural analogs used under the optimized conditions. Phallacidin and phallisacin were used as negative controls, and skimmed milk was used as the blank control for the specificity study. The mixture of the five toxins was tested.

Different α-amanitin concentrations were assayed and linear curves were established to investigate the sensitivity of the method. Under the optimized conditions, the third step was changed to add α-amanitin at final concentrations of 0, 0.1, 0.25, 0.5, 1, 2, 4, and 6 μg/mL and incubated at 37 °C for 2 h.

### 4.7. Preparation of the Samples

In order to assess the effect of the matrix on the assay, both mushroom and urine matrices were used for the determination of the sensitivity of α-amanitin. The standards of α-amanitin with the same concentration (0, 0.1, 0.25, 0.5, 1, 2, 4, and 6 μg/mL) as the standard curve were added to the normal urine and mushroom leachate, respectively, and the concentration of α-amanitin in the spiked samples was calculated with reference to the standard curve, and the recoveries were calculated. Three replicates of the experiment were performed for each group. The spiked samples were detected using a commercial kit for α-amanitin, and the experimental procedures were carried out in accordance with the instructions. The difference in recovery between MELISA and the ELISA kit was compared.

To further verify the feasibility of the method for detecting the compounds in actual samples, mushroom leachate and urine were tested using the method, and the results were compared with the national standard of high-performance liquid chromatography (HPLC). Normal human urine was used as the negative control and skimmed milk was the blank control in the experiment. The samples were pretreated as follows: one-third of a cap of an amatoxin sample was taken in a mortar, 8 mL of water was added for grinding, and the supernatant was centrifuged; the total volume of the supernatant was 5.7 mL. The processed amatoxin leachate was subsequently removed for gradient dilution and set aside. Skim milk, the mushroom leachate, a urine sample from a poisoned patient, and a normal urine sample was added and incubated in 20 μL, which was brought up to 100 μL with 5% skim milk and incubated at 37 °C for 2 h.

### 4.8. Molecular Docking Simulation

Molecular docking has been widely used to explore and model how ligand and receptor molecules recognize each other. Molecular docking simulation focuses on the interaction between two molecules to predict the simulated binding mechanism [55,56]. In this study, molecular docking simulations were performed to further understand the recognition sites between the aptamer and α-amanitin. First, the 3D structure of the aptamer was obtained from online modeling. Then, simulation operations were performed using Autodock software.

## Figures and Tables

**Figure 1 molecules-27-00538-f001:**
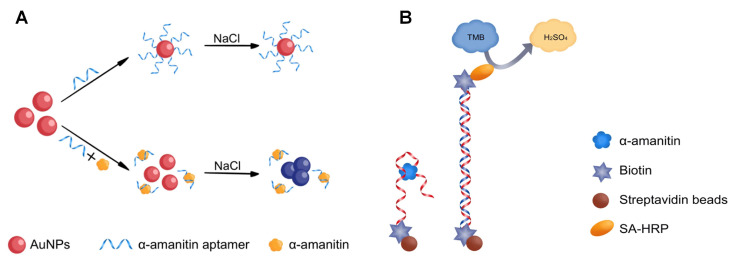
Experimental schematic. (**A**) Schematic illustration of the affinity of the α-amanitin truncated aptamer by colloidal gold spectrophotometry. (**B**) Schematic illustration based on MELISA to detect α-amanitin.

**Figure 2 molecules-27-00538-f002:**
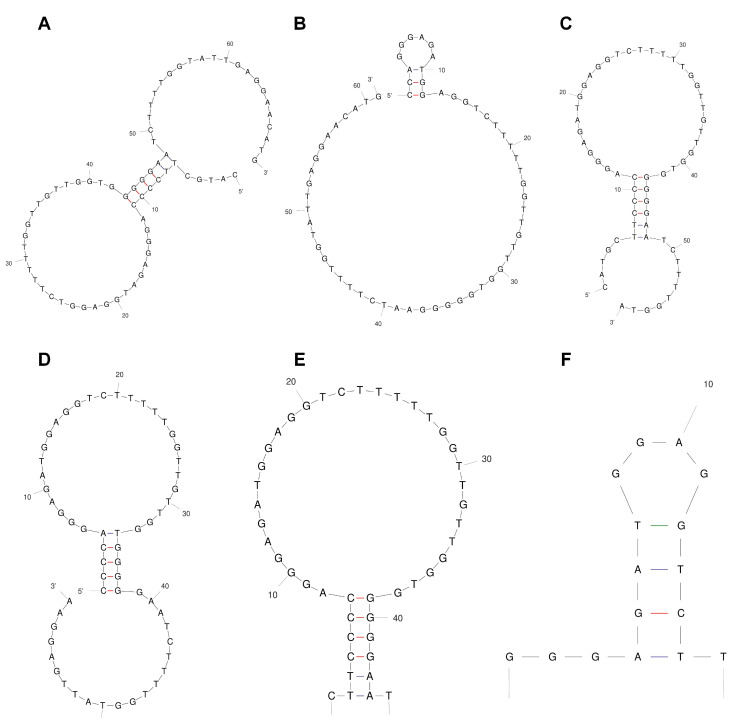
Secondary structures of the original α-amanitin aptamer and the truncated aptamer. The secondary structures of the aptamers were predicted and analyzed using Mfold software. (**A**–**F**) are Apt, Apt-2, Apt-8, Apt-10, Apt-11, and Apt-12.

**Figure 3 molecules-27-00538-f003:**
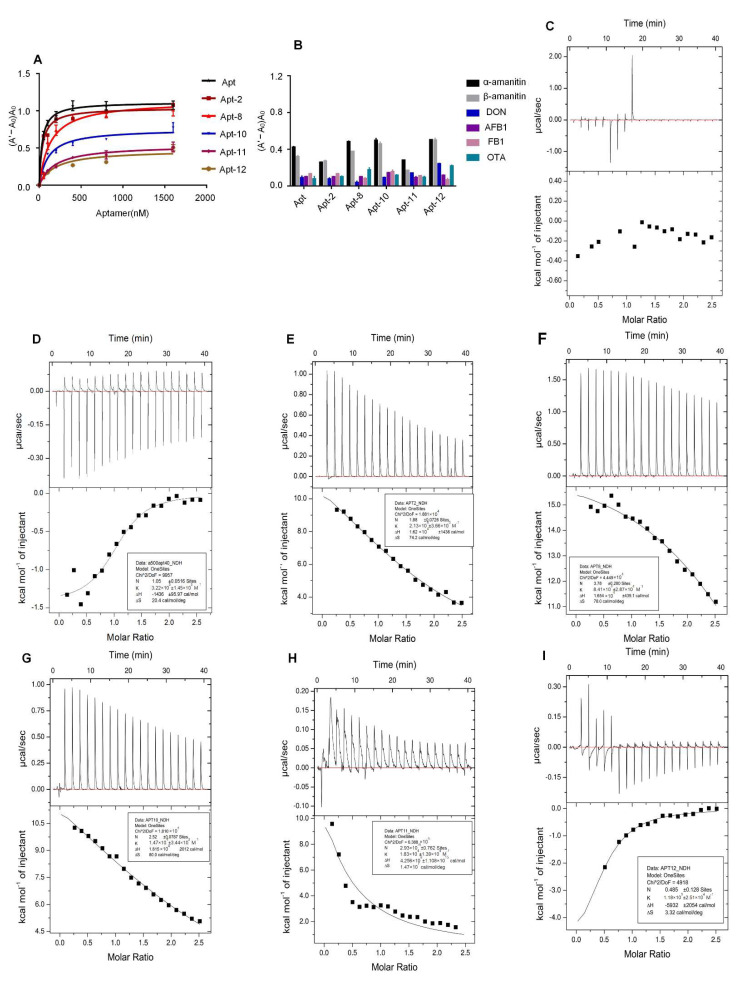
Haracterization of the α-amanitin aptamer and truncated aptamer. (**A**) Determining α-amanitin aptamer and truncated aptamer affinity. The aptamer concentrations (0, 50, 100, 200, 400, 800, and 1600 nM) were combined with equal amounts of α-amanitin (1 μg/mL), and the Kd values were analyzed by non-linear regression analysis using GraphPad Prism 8 software. (**B**) Specific test of aptamers with α-amanitin and other non-targets (β-amanitin, DON, AFB1, FB1, OTA). (**C**) Blank control: ITC results for titration of α-amanitin with ultrapure water. (**D**–**I**) ITC results of aptamer titration of α-amanitin.

**Figure 4 molecules-27-00538-f004:**
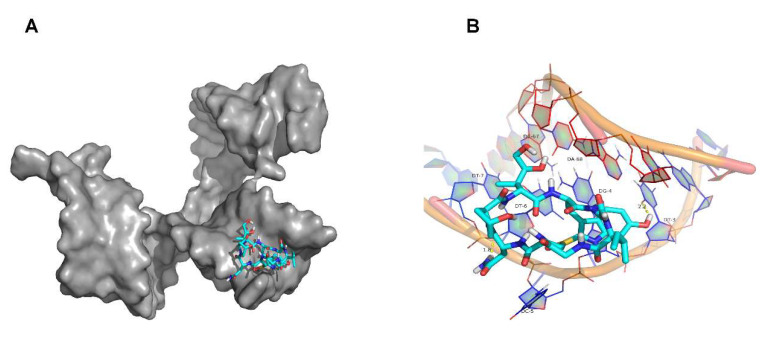
The interactive mechanism between the aptamer and α-amanitin was investigated by the molecular docking simulation technique. (**A**) Schematic diagram of α-amanitin binding to the aptamer (autodock mapping). (**B**) Base sites of the interaction between α-amanitin and the aptamer (pymol mapping).

**Figure 5 molecules-27-00538-f005:**
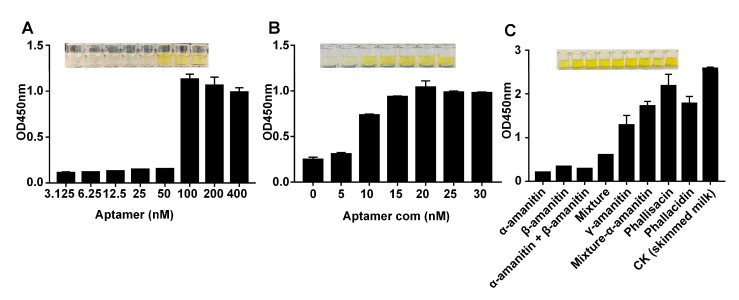
Optimizing the MELISA to detect α-amanitin. (**A**) Optimizing the aptamer concentration. The aptamer concentrations were 3.125, 6.25, 12.5, 25, 50, 100, 200, and 400 nM, in that order. (**B**) Optimizing the complementary chain concentrations. The complementary chain concentrations were 0, 5, 10, 15, 20, 25, and 30 nM, in that order. (**C**) Specificity of the experimental methods. Method specificity was determined with α-amanitin and other non-target substances, including β-amanitin, γ-amanitin, phallacidin, and phallisacin.

**Figure 6 molecules-27-00538-f006:**
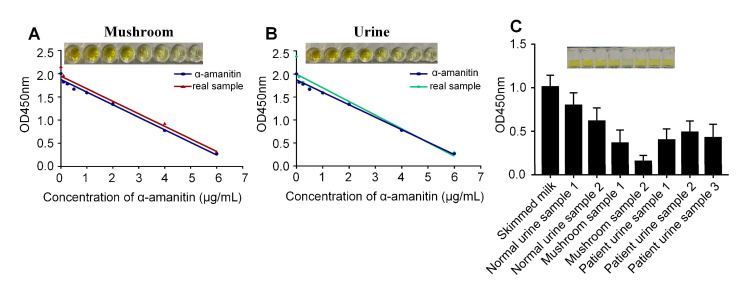
Experimental method validation. (**A**) Sensitivity detection. The concentrations of α-amanitin in standard and mushroom were 0, 0.1, 0.25, 0.5, 1, 2, 4, and 6 μg/mL, and the corresponding OD450 values were used to prepare the standard curve. Standard curve of α-amanitin in mushroom: y = −0.2701x + 1.945(R^2^ = 0.969). (**B**) The concentrations of α-amanitin in standard and urine were 0, 0.1, 0.25, 0.5, 1, 2, 4, and 6 μg/mL, and the corresponding OD450 values were used to prepare the standard curve. Standard curve of α-amanitin in urine: y = −0.2969x + 1.996(R^2^ = 0.94). (**C**) Determination of α-amanitin in the actual samples. Samples (mushroom leachate and urine) were tested with skim milk as a blank control.

**Figure 7 molecules-27-00538-f007:**
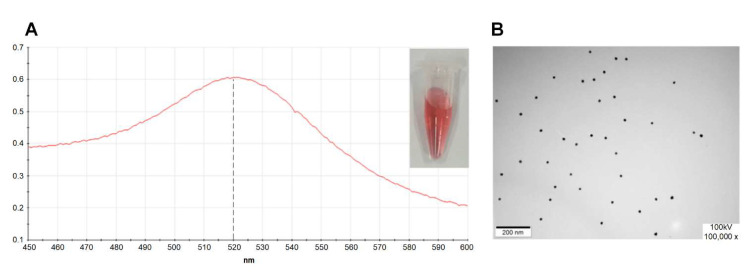
Characterization of colloidal gold. (**A**) UV-visible absorption spectra of the AuNPs. (**B**) TEM image of the AuNPs.

**Table 1 molecules-27-00538-t001:** Primitive and truncated aptamer sequences (5′-3′), dissociation constants (Kd), and dG values.

Name	Sequence (5′-3′)	Kd (Nm)	dG
Apt [39]	CATGCTTCCCCAGGGAGATGGAGGTCTTTTTGGTTGTTGGTGGGGGAATCTTTTGGTATTGAGGAACATG	33.6 ± 3.928	−3.13
Apt-2	CCAGGGAGATGGAGGTCTTTTTGGTTGTTGGTGGGGGAATCTTTTGGTATTGAGGAACATG	37.9 ± 6.053	−1.16
Apt-8	CATGCTTCCCCAGGGAGATGGAGGTCTTTTTGGTTGTTGGTGGGGGAATCTTTTGGTA	130.9 ± 31.1	−3.13
Apt-10	CCCCAGGGAGATGGAGGTCTTTTTGGTTGTTGGTGGGGGAATCTTTTGGTATTGAGGAA	108.7 ± 18.23	−2.41
Apt-11	CTTCCCCAGGGAGATGGAGGTCTTTTTGGTTGTTGGTGGGGGAAT	184.4 ± 37.08	−3.13
Apt-12	GGGAGATGGAGGTCTT	185.2 ± 60.87	−0.25

**Table 2 molecules-27-00538-t002:** ITC results for original aptamer and truncated aptamer.

Aptamer	N (Sites)	Kd (μM)	ΔH (kcal/mol)	ΔG (kcal/mol)	−TΔS (kcal/mol)
Apt	1.05	3.1	−1.436	−7.515	−6.079
Apt-2	1.88	46.9	16.21	−5.36	−2.157
Apt-8	3.78	11.9	16.54	−2.7	−23.24
Apt-10	2.52	68	18.15	5.69	−23.84
Apt-11	2.93 × 10^4^	54.6	4.256 × 10^6^	−0.124 × 10^6^	4.38 × 10^6^
Apt-12	0.485	84.7	−5.932	−6.921	0.989

**Table 3 molecules-27-00538-t003:** The detected concentrations of α-amanitin were compared with the standards added to the actual samples.

Samples	Added Concentration (ng/mL)	Kit Measured (ng/mL)	Recovery (%)	MELISA (ng/mL)	Recovery (%)
Mushroom	10	8.413	84.13	8.951	89.51
	50	47.14	94.28	42.85	85.7
	100	90.48	90.48	88.16	88.16
	200	191.3	95.65	194.8	97.4
Urine	10	8.304	83.04	8.94	89.4
	50	46.46	92.92	43.08	86.1
	100	90.21	90.21	90.16	90.16
	200	192.93	96.46	192.6	96.3

**Table 4 molecules-27-00538-t004:** Comparison of two detection methods.

Sample	Magnetic Bead-ELISA (*n* = 2)	HPLC (*n* = 2)
Normal urine sample1	−−	−−
Normal urine sample2	−−	−−
Mushroom sample1	++	++
Mushroom sample2	++	++
Patient urine sample1	++	++
Patient urine sample2	++	++
Patient urine sample3	++	++

## Data Availability

The data presented in this study are available in Appendix A.

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
