# Peer review of "Utilizing the DNA Aptamer to Determine Lethal α-Amanitin in Mushroom Samples and Urine by Magnetic Bead-ELISA (MELISA)"

_molecules, 2022, doi:10.3390/molecules27020538_

Round 1
Reviewer 1 Report
This paper is ready to be published
Author Response
Point 1: This paper is ready to be published.
Response 1: We thank you for taking the time to review our manuscript and for your recognition of our manuscript.
Reviewer 2 Report
Accept in present form.
Author Response
Point 1: Accept in present form.
Response 1: We thank you for taking the time to review our manuscript and for your recognition of our manuscript.
Reviewer 3 Report
Although the manuscript describes a novel aptameric biosensor based approach for detection of amanitin from mushroom samples and urine, it lacks coherent and lucid explanation of results. Moreover, a stronger Introduction with inclusion of latest literature is suggested and this must be made conspicuous in discussion as well. I recommend the authors to include relevant studies ( https://doi.org/10.1016/j.jhazmat.2021.127939 and https://doi.org/10.3390/ijms221910846 ) to bolsters introduction and discussion.
-The impact of truncation of aptamer is not described properly. What happens to sensitivity and specificity of aptamer? Does it improve or worsen? In above suggested paper--authors prove the truncation in stem loop do not affect the aptamer performance, however, point mutations in loops (major or minor) reduces binding of the target to aptamer. Therefore, I suggest authors to discuss this point in details and derive your conclusion supported by experimental data.
-I quite fail to correlate the objective of analysis of mushroom in relation to urine samples. Why urine?
-The descriptive writing lacks uniformity in use of terminology. Please be consistent and coherent.
-A lucid discussion of mutation in aptamer is essential to show significance of the study.
Overall, the study have scientific merit and can be considered for publication after incorporating all suggested comments.
Author Response
Thank you for your review. Please see the attachment.

This manuscript is a resubmission of an earlier submission. The following is a list of the peer review reports and author responses from that submission.
Round 1
Reviewer 1 Report
This manuscript reported truncation work on previously reported amanitin aptamer sequence and proposed an aptamer-conjugated magnetic bead-based ELISA method for the amanitin detection. The original aptamer was truncated into five sequences which then undergone specificity and affinity analysis. However, these five truncated aptamers were reported to have poor specificity and affinity to amanitin compared to original aptamer, and thus the original aptamer was further used in this study for the detection purpose. First, failed attempt on producing good truncated aptamer as stated by the authors in the manuscript as “poor” and “worse”should not be included in manuscript. Second, results from figure 3A and 3B are actually showing that the truncated aptamer 8 and 10 are having almost similar sensitivity compared to the original aptamer, thus the conclusion is not well supported. On the other hand, the detection method shows low novelty and the limit of detection given must be determined correctly, not by the lowest range tested.
Reviewer 2 Report
This study developed a MELISA-based method for the detection of α-amanitin through the recognition of aptamer. However, the aptamer for specifically identifying α-amanitin has been reported by previous studies (e.g., Acta Biochim Pol. 2017, 64, 401-40). From the point of view of developing a new aptamer, the novelty of this work still has room to be improved. Additionally, the earlier published literature has proposed the gold nanoparticle- and magnetic nanoparticle-based detection strategy. Thus, I recommend that this paper is not suitable to be published in Molecules.
Comments
1) The authors did not show how to determine the Kd and dG values between aptamers and the target analytes. Would you please show the equation for calculating the Kd and dG values?
2) Please suggest the possible approach to improve the selectivity of the proposed probe toward α-amanitin in the presence of β-amanitin.
3) The authors did not use the proposed method to determine the level of α-amanitin in real samples.
Reviewer 3 Report
-Authors are advised to rewrite the title as “Utilizing the DNA Aptamer to Determine Lethal α-Amanitin in in mushroom samples and urine by Magnetic Bead-ELISA (MELISA)”
-Authors have described that “The detection limit was 0.1 μg/mL; linear in the range of 0.01–6 μg/mL and was used to detect α-amanitin in mushroom samples and urine” the method linearity is was studied 0.01–6 μg/mL, then how the obtained detection limit is higher than the lower value of the calibration concentration. Explanation required.
-Authors should also describe the detection limits of both mushroom samples and urine separately, however the matrices of studied samples are quite different.
-Keywords should also include the name of studied samples.
-Please add regulation if available “α-Amanitin is the main lethal toxin in amanita, and the human lethal dose is about 0.1 mg/kg”.
-Check all abbreviations in the main body of the text. Use either HPLC or LC,
-check chromatography-mass spectroscopy (LC-MS). Use correct name.